# Fetal influence on the human brain through the lifespan

Kristine B Walhovd[1,2]*, Stine K Krogsrud[1], Inge K Amlien[1], Øystein Sørensen[1], Yunpeng Wang[1], Anne Cecilie S Bråthen[1], Knut Overbye[1], Jonas Kransberg[1], Athanasia M Mowinckel[1], Fredrik Magnussen[1], Martine Herud[1], Asta K Håberg[3], Anders Martin Fjell[1,2], Didac Vidal-Pineiro[1]

[1]Center for Lifespan Changes in Brain and Cognition, University of Oslo, Oslo, Norway; [2]Computational Radiology and Artificial Intelligence, Department of Radiology and Nuclear Medicine, Oslo University Hospital, Oslo, Norway; [3]Department of Neuromedicine and Movement Science, Faculty of Medicine and Health Sciences, Norwegian University of Science and Technology, Oslo, Norway

**Abstract** Human fetal development has been associated with brain health at later stages. It is unknown whether growth in utero, as indexed by birth weight (BW), relates consistently to lifespan brain characteristics and changes, and to what extent these influences are of a genetic or environmental nature. Here we show remarkably stable and lifelong positive associations between BW and cortical surface area and volume across and within developmental, aging and lifespan longitudinal samples (N = 5794, 4–82 y of age, w/386 monozygotic twins, followed for up to 8.3 y w/12,088 brain MRIs). In contrast, no consistent effect of BW on brain changes was observed. Partly environmental effects were indicated by analysis of twin BW discordance. In conclusion, the influence of prenatal growth on cortical topography is stable and reliable through the lifespan. This early-life factor appears to influence the brain by association of brain reserve, rather than brain maintenance. Thus, fetal influences appear omnipresent in the spacetime of the human brain throughout the human lifespan. Optimizing fetal growth may increase brain reserve for life, also in aging.

## eLife assessment

This **valuable** study uses multiple large neuroimaging datasets acquired at different points through the lifespan to provide **solid** evidence that birth weight (BW) is associated with robust and persistent variations in cortical anatomy, but less-substantial influences on cortical change over time. These findings, supported by robust statistical methods, illustrate the long temporal reach of early developmental influences and carry relevance for how we conceptualize, study, and potentially modify such influences more generally. The article will be of interest to people interested in brain development and aging.

## Introduction

It is established that a substantial portion of functional variation through the lifespan, including in older age, is of neurodevelopmental origin (*Kovacs et al., 2014*; *Fjell et al., 2015*; *Bale, 2015*; *Muller et al., 2014*). Evidence converges on early-life factors being important for normal individual differences in brain, mental health, and cognition across the lifespan (*Walhovd et al., 2016*; *Walhovd et al., 2012*; *Vidal-Pineiro et al., 2021*; *Dooley et al., 2022*; *Elliott et al., 2021*), as well as risk of psychiatric (*Anderson et al., 2021*) and neurodegenerative disease in older age (*Tuovinen et al., 2013*). Obtaining reliable indicators of individual early-life factors is a major challenge. In this regard,

*For correspondence:
k.b.walhovd@psykologi.uio.no

Competing interest: The authors declare that no competing interests exist.

birth weight (BW) stands out as a solid available measure. BW reflects fetal and maternal genetic, but also other in utero environmental factors affecting fetal growth (*Beaumont et al., 2018*; *Li et al., 2021*; *Willis et al., 2022*), including brain growth (*Walhovd et al., 2012*; *Raznahan et al., 2012*; *Casey et al., 2017*; *Halevy et al., 2021*). By now, a series of studies have established that BW relates positively to mental health, cognitive function, and brain characteristics, including neuroanatomical volumes and cortical surface area as measured in different age groups (*Muller et al., 2014*; *Walhovd et al., 2016*; *Walhovd et al., 2012*; *Raznahan et al., 2012*; *Wheater et al., 2021*; *Eriksson et al., 2000*). However, it is unknown whether and how BW relates to brain characteristics through the lifespan, how consistent effects are within and across samples, whether BW is associated with lifespan brain changes, and to what extent lifespan effects of BW on the brain are of an environmental, rather than genetic nature. We address these questions, which are critical to understand how and when the human brain can be influenced through the lifespan, in the present study. On an overarching level, this study also addresses current debates in the field of lifespan cognitive neuroscience, namely (1) whether consistent, reproducible relationships between phenotypes relevant for mental health and function and inter-individual differences in brain characteristics can be found (*Marek et al., 2022*), and (2) to what extent the effects found in and ascribed to brain aging may actually reflect early-life influences, rather than longitudinal changes in older age (*Walhovd et al., 2016*; *Vidal-Pineiro et al., 2021*; *Walhovd et al., 2020*; *Walhovd et al., 2023*).

There are at least two different ways by which the effects of fetal growth, as indexed by BW, could work to produce the brain effects observed so far. (1) In line with a brain reserve model (*Katzman et al., 1988*; *Schofield et al., 1997*), higher BW could be associated with greater brain growth before birth. This seems likely, given that the effects are seen also in young populations (*Walhovd et al., 2012*; *Raznahan et al., 2012*). However, from the so far largely cross-sectional, or mixed models, several questions remain unanswered: Is this a fixed effect at the time of birth? Does higher BW also have carry-over effects to greater development in childhood and adolescence? In line with a brain maintenance model (*Nyberg et al., 2012*), is higher BW associated with better maintenance of brain volumes in the face of age-related changes in older adulthood? While effects are found in young populations (*Walhovd et al., 2016*; *Walhovd et al., 2012*; *Raznahan et al., 2012*), reduced atrophy in aging is a possible additional effect of higher BW that should be investigated, given the known relationships between birth size and brain volumes also in older age (*Muller et al., 2014*). The possible effects of BW on later brain development and brain maintenance in adulthood can only be investigated by longitudinal brain imaging spanning all stages of human life. Furthermore, as BW normally reflects both genetic and prenatal environmental factors, and an environmental BW contribution to brain differences has been shown in young monozygotic (MZ) twins (*Raznahan et al., 2012*; *Casey et al., 2017*; *Halevy et al., 2021*), we need to study brain effects of BW discordance in MZ twins in this context to disentangle possible non-genetic contributions of BW through the lifespan.

We hypothesized that there are persistent effects of BW on brain characteristics through the lifespan, and hence, that these would be consistent within and across samples of varying age and origin. We test this in a Norwegian sample covering the lifespan (Lifespan Changes in Brain and Cognition [LCBC]) (*Walhovd et al., 2016*; *Walhovd et al., 2020*), the US developmental sample Adolescent Brain Cognitive Development (ABCD) (*Casey et al., 2018*; *Garavan et al., 2018*), and the older adult UK Biobank (UKB) (*Alfaro-Almagro et al., 2018*; *Nobis et al., 2019*) sample. The associations of BW and cortical surface, thickness, volume, and their change were investigated vertex-wise in a total of 5794 persons (of whom 5718 with repeated scans and 386 MZ twins) with 12,088 longitudinal observations, 4–82 y of age at baseline, followed for up to ~8.3 y. Based on previous results (*Walhovd et al., 2016*; *Walhovd et al., 2012*; *Raznahan et al., 2012*), we hypothesized such effects to be driven primarily by positive associations between BW and cortical area, with lesser, if any, effects on cortical thickness. We expected positive effects on cortical volume corresponding to positive effects on cortical area. We hypothesized that effects would be stable, so that BW mainly affects the brain 'intercept' and does not relate much to brain changes. That is, we hypothesize a threshold model, whereby higher BW yields greater cortical area, and hence cortical volume, to begin with, rather than a maintenance model, whereby higher BW serves to protect against atrophy in aging. Moreover, we hypothesized that effects could not be explained solely by genetics, so that BW discordance in a subsample of MZ twins would also result in differences in brain characteristics through the lifespan.

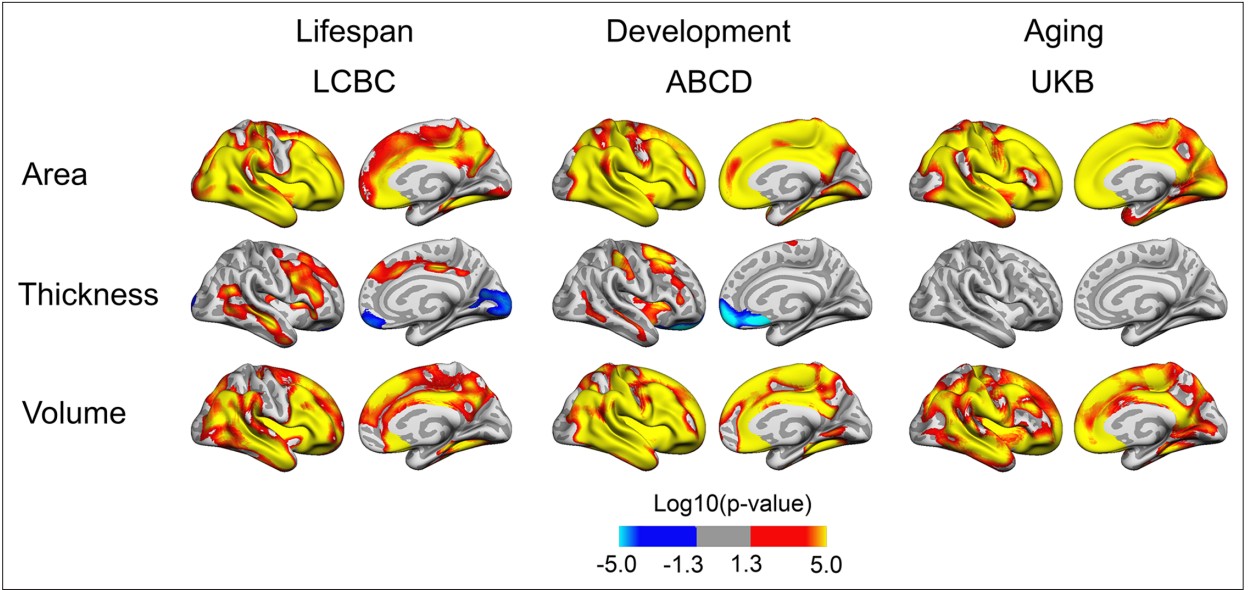

**Figure 1.** Relationships of birth weight and cortical characteristics across Lifespan Changes in Brain and Cognition (LCBC), Adolescent Brain Cognitive Development (ABCD), and UK Biobank (UKB) samples. Age, sex, time (interval since baseline), and scanner site (as well as ethnicity in the ABCD) were controlled for. Significant relationships are shown for area, thickness, and volume for each sample, from left to right: lateral view and medial view, right hemisphere.

The online version of this article includes the following figure supplement(s) for figure 1:

**Figure supplement 1.** Relationships of birth weight and cortical characteristics in both hemispheres across Lifespan Changes in Brain and Cognition (LCBC), Adolescent Brain Cognitive Development (ABCD), and UK Biobank (UKB) samples when controlling for age, sex, time (interval since baseline), and scanner site (as well as ethnicity in the ABCD).

**Figure supplement 2.** Relationships of birth weight and cortical characteristics across Lifespan Changes in Brain and Cognition (LCBC), Adolescent Brain Cognitive Development (ABCD), and UK Biobank (UKB) samples when controlling for education, age, sex, time (interval since baseline), and scanner site (as well as ethnicity in the ABCD).

**Figure supplement 3.** Relationships of birth weight and cortical characteristics across Lifespan Changes in Brain and Cognition (LCBC) and Adolescent Brain Cognitive Development (ABCD) samples when controlling for gestational length in weeks (LCBC) or weeks born prematurely (ABCD), age, sex, time (interval since baseline), and scanner site (as well as ethnicity in the ABCD).

**Figure supplement 4.** Relationships of birth weight and cortical characteristics across Lifespan Changes in Brain and Cognition (LCBC), Adolescent Brain Cognitive Development (ABCD), and UK Biobank (UKB) when restricting the samples to participants with birth weights between 2.5 and 5.0 kg, controlling for age, sex, time (interval since baseline), and scanner site (as well as ethnicity in the ABCD).

**Figure supplement 5.** Relationships of birth weight and cortical characteristics across Lifespan Changes in Brain and Cognition (LCBC), Adolescent Brain Cognitive Development (ABCD), and UK Biobank (UKB) samples when controlling for intracranial volume (ICV), age, sex, time (interval since baseline), and scanner site (as well as ethnicity in the ABCD).

**Figure supplement 6.** Comparison between spline (generalized additive mixed model [GAMM]) and linear (linear mixed model [LME]) models on the effect of birth weight on cortical characteristics.

## Results

Cortical surfaces were reconstructed from T1-weighted anatomical MRIs by use of FreeSurfer v6.0 (LCBC and UKB) and 7.1. (ABCD) (https://surfer.nmr.mgh.harvard.edu/; *Dale et al., 1999*; *Fischl et al., 2002*; *Reuter et al., 2012*; *Jovicich et al., 2013*), yielding maps of cortical area, thickness, and volume. Vertex-wise analyses were run with spatiotemporal linear mixed effects modeling (FreeSurfer v6.0.0 ST-MLE package) to assess regional variation in the relationships between BW and cortical structure and its change. All analyses were run with baseline age, sex, scanner site, and time (scan interval) as covariates. For ABCD specifically, ethnicity was also included as a covariate. For consistency of multiple comparison corrections across analyses, the results were thresholded at a cluster-forming threshold of 2.0, $p < 0.01$, with a cluster-wise probability of $p < 0.0025$ ($p < .05/2$ hemispheres).

## The lifespan relationship of BW and cortical volume, surface area, and thickness

Associations of BW and cortical characteristics are shown in *Figure 1* (for the right hemisphere) and in *Figure 1—figure supplement 1* (for both hemispheres). Across all cohorts, widespread positive associations were observed between BW and cortical area. These were highly consistent across lifespan (LCBC), developmental (ABCD), and aging (UKB) cohorts, and there were bilateral overlapping effects across most of the cortical mantle. As expected, BW had, in general, lesser effects on cortical thickness and no significant effects on thickness were observed in the UKB. There were, however, some lateral-positive and medial-negative effects in the LCBC and ABCD cohorts. We note that corresponding effects with increased medial frontal and occipital cortical thickness have been found associated with white matter alterations (reduced FA) in young adults born preterm with very low BW compared to term-born controls (*Rimol et al., 2019*). BW was significantly positively associated with cortical volume across much of the cortical mantle. In sum, broad, bilateral, positive associations were observed across cohorts for cortical area and volume.

Additionally controlling for education level had little effect on results (see *Figure 1—figure supplement 2*). Information on gestational length (i.e., whether there was premature birth) was not available for all participants. Importantly, this information was lacking for the older participants, that is, this information is not available in detail for UKB, and since this information for the LCBC was drawn from the Medical Birth Registry of Norway (MBRN), only established in 1967, this was not available for the older part of the LCBC sample either. The majority of the LCBC sample and the ABCD sample had information on gestational length, however (LCBC: n = 514; gestational length in weeks: M = 40.0 wk, SD = 1.9, range = 25–44; ABCD: n = 3306; weeks premature: M = 1.0 weeks premature, SD = 2.1, range = 0.0–13.0). Controlling for gestational age in these subsamples had relatively little effect on results (see *Figure 1—figure supplement 3*). However, as expected with reduced power in the LCBC sample, the effects in this analysis were somewhat narrower. Effects in ABCD, where almost all participants were retained for analysis, showed no sign of decrease with control for gestational length. When restricting all samples to participants with BW between 2.5 and 5.0 kg, results were also very similar (see *Figure 1—figure supplement 4*). As expected from the widespread effects on cortical area and volume, effects were partly generic, with analyses controlling for intracranial volume (ICV) showing more restricted effects (see *Figure 1—figure supplement 5*). However, consistent significant positive effects of BW on cortical area also when controlling for ICV were observed across all three cohorts in lateral temporal and frontal areas.

## BW effects on cortical change

To test the effect of BW on cortical change, we reran the analyses with BW × time and age × time interactions. Note that BW × time (i.e., within-subject follow-up time) represents the contrasts of interest while age – and age interactions – is used to account for the differences in age across individuals. Significant BW × time interactions on cortical characteristics were observed in restricted and non-overlapping regions across samples (see *Figure 2* [depicting right hemisphere results]; for visualization of effects in both hemispheres, see *Figure 2—figure supplement 1*). Per direction of effect, the effect of BW differences was apparently reduced over time for area in LCBC and ABCD, whereas no interaction effects on area were significant in UKB. A mixture of positive (ABCD) and negative (LCBC, UKB) interaction effects were significant for thickness and volume.

Visualization of the interaction effects as seen in *Figure 2* and *Figure 2—figure supplement 1*, by splitting the sample into two based on BW, did not yield convincing evidence for these interactions, as shown in *Figure 2—figure supplements 2 and 4*. In plots of LCBC data, where the number of follow-ups varied, and a select portion had longer follow-up, it appeared that the effect of BW was reduced over time in these restricted regions. However, virtually parallel trajectories for the ABCD and UKB subsamples with lower and higher BW suggested that the effect size even within the areas of significant interactions of BW and time was negligible. Since the UKB and ABCD samples here consisted of samples having two time points only, whereas LCBC consisted of a mix of number of follow-ups over a longer time period, there might be sample-specific selection effects also regarding other characteristics than BW that can influence these effects in LCBC. For instance, participants who do not drop out tend to have better health, cognitive ability, and education, which again may relate

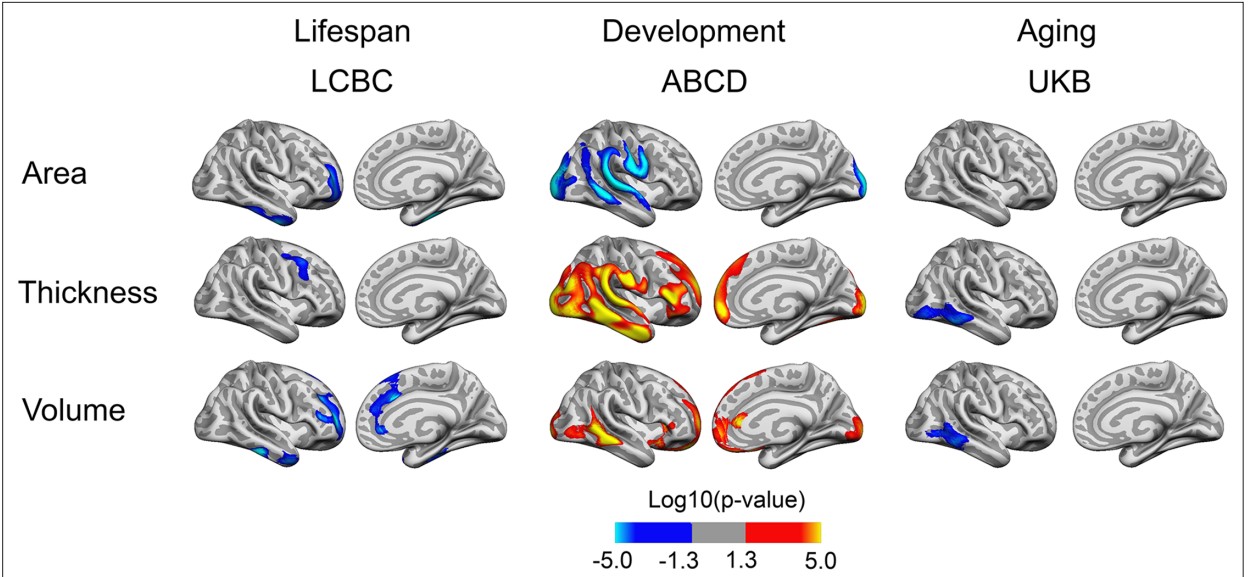

**Figure 2.** Interactions of birth weight (BW) and time on cortical characteristics across Lifespan Changes in Brain and Cognition (LCBC), Adolescent Brain Cognitive Development (ABCD), and UK Biobank (UKB) samples. Age, sex, scanner site, time, and BW (as well as ethnicity in the ABCD) were controlled for. Significant relationships are shown, from left to right: lateral view, right and left hemisphere; and medial view, right and left hemisphere.

The online version of this article includes the following figure supplement(s) for figure 2:

**Figure supplement 1.** Interactions of birth weight (BW) and time on cortical characteristics in both hemispheres across Lifespan Changes in Brain and Cognition (LCBC), Adolescent Brain Cognitive Development (ABCD), and UK Biobank (UKB) samples when controlling for age, sex, scanner site, time, birth weight, and the interaction of baseline age and time (as well as ethnicity in the ABCD).

**Figure supplement 2.** Plots showing individual data points and expected trajectories for cortical area within the significant regions (refer to *Figures 1 and 2*) of each sample split into two based on birth weight (BW) (higher BW in red color = upper half, lower BW in blue color = lower half of BW distribution).

**Figure supplement 3.** Plots showing individual data points and expected trajectories for cortical thickness within the significant regions (refer to *Figures 1 and 2*) of each sample split into two based on birth weight (BW) (higher BW in red color = upper half, lower BW in blue color = lower half of BW distribution).

**Figure supplement 4.** Plots showing individual data points and expected trajectories for cortical volume within the significant regions (refer to *Figures 1 and 2*) of each sample split into two based on birth weight (BW) (higher BW in red color = upper half, lower BW in blue color = lower half of BW distribution).

**Figure supplement 5.** The degree of within-sample replicability of birth weight (BW) effects on cortical structure for Lifespan Changes in Brain and Cognition (LCBC), Adolescent Brain Cognitive Development (ABCD), and UK Biobank (UKB).

**Figure supplement 6.** Comparison between spline (generalized additive mixed model [GAMM]) and linear (linear mixed model [LME]) models on the effect of birth weight on cortical change.

positively to the brain measures studied here (*Nyberg et al., 2010*; *Wolke et al., 2009*). Thus, caution is advised in interpreting effects seen only with longer follow-up in the LCBC sample.

Both BW effects on cortical characteristics and cortical change were rerun (ROI-wise) using spline models that accounted for possible nonlinear effects of age on cortical structure. The results were comparable to those reported in *Figures 1 and 2*. See *Figure 1—figure supplement 6* and *Figure 2— figure supplement 6* for BW effects on cortical characteristics and cortical change, respectively.

## Consistency of spatial relationships across and within samples

Next, we assessed whether the cortical correlates of BW (βeta-maps) showed a similar topographic pattern across the three independent datasets (UKB, ABCD, and LCBC). The results showed that all the spatial comparisons were statistically significant (p<0.05, FDR-corrected). That is, the topography of the effects of BW on cortical structure was comparable across datasets – the pairwise spatial correlation of a given cortical correlate of BW (e.g., BW effects on cortical area) was similar when estimated from two different datasets. The spatial correlations were the highest for the volume measures (*r* = 0.64–0.79), and overall also high (*r* = 0.51–0.71) for area measures, whereas for cortical thickness,

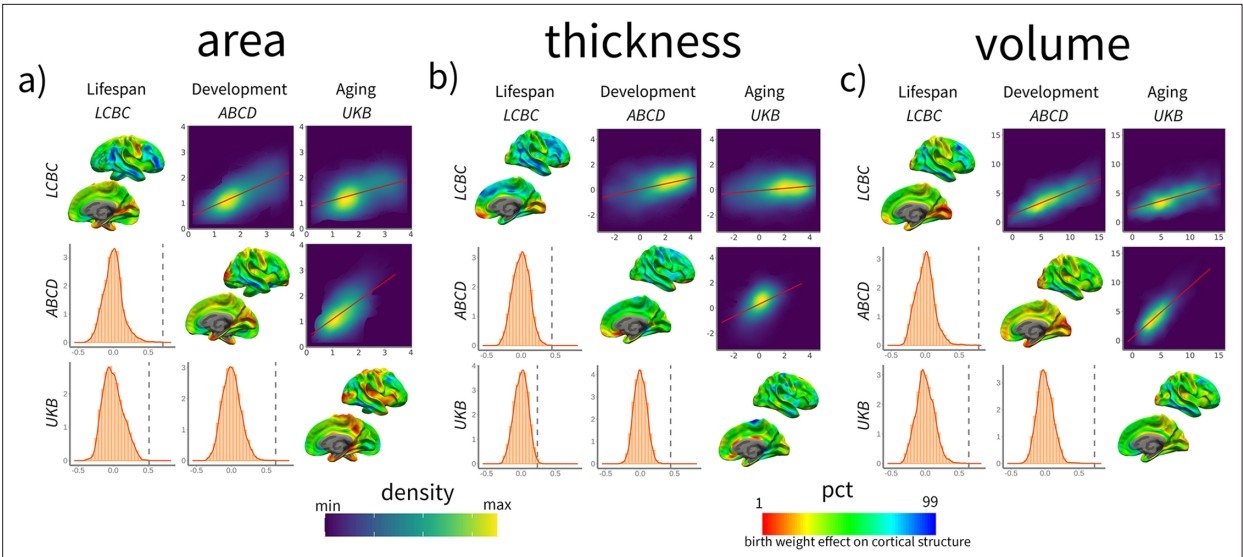

**Figure 3.** Spatial correlation of birth weight (BW) effects on brain structure across datasets for cortical (**a**) area, (**b**) thickness, and (**c**) volume. Spatial correlation of BW effects on brain structure across datasets. For each panel, the upper triangular matrix shows Pearson's (*r*) pairwise spatial correlation between the different cohorts' cortical maps. Data is shown as a color-density plot. The red line represents the fitting between the two maps. The lower triangular matrix shows the significance testing. The dashed-gray line shows the empirical correlation, while the orange histogram represents the null distribution based on the spin test. The diagonal shows the effect of BW on cortical structure (right hemisphere shown only). Note that the βeta-maps are shown as a percentile red–green–blue scale, where red represents a lower (or more negative) effect of BW on cortical structure and vice versa. Units in the density maps represent BW effects as mm/g, mm²/g, and mm³/g (10e⁻⁵) for cortical thickness, area, and volume, respectively.

The online version of this article includes the following figure supplement(s) for figure 3:

**Figure supplement 1.** Spatial correlation of birth weight effects on brain structure change across datasets.

they were more moderate (*r* = 0.24–0.45). See spatial correlations for the right hemisphere cortical volume in *Figure 3* and the full model summary in *Supplementary file 1*. The results were qualitatively comparable when using -log₁₀ (p) significance values instead of βeta estimates, as shown in *Supplementary file 1*. The same pattern of results was largely seen also for spatial correlation of the maps capturing BW-associated cortical characteristics when controlling for ICV. The correlations were then on average somewhat lower, but there were still only significant positive correlations across LCBC, ABCD, and UKB (see *Supplementary file 1*).

In contrast, the spatial correlation of the maps capturing BW-associated cortical *change* (i.e., BW × time contrast) was either unrelated (n = 7) or showed negative associations between cohorts (n = 2). The spatial correlations of BW on cortical change were *r* = –35 to –0.05 for area, *r* = –0.35 to –0.08 for volume, and *r* = –0.20 to –0.04 for thickness. See a visual representation in *Figure 3—figure supplement 1* and full stats in *Supplementary file 1*.

In sum, the spatial correlation analyses imply that the different datasets show a comparable topography of BW effects across the cortical mantle; that is, the areas more and less affected by BW were common across datasets. Thus, the BW effects on cortical structure are robust and replicable across very different datasets. In contrast, the effects of BW on cortical change are not robust across datasets, showing dissimilar topographies.

Additionally, we performed replicability analyses both across and within samples to further investigate the robustness of the effects of BW on cortical characteristics and cortical change. Split-half analyses within datasets were performed to investigate the replicability of significant effects (*Kharabian Masouleh et al., 2019*; *Open Science, 2015*) of BW on cortical characteristics within samples (refer to *Figure 1*). These analyses further confirmed that the significant effects were largely replicable for volume and area, but not for thickness (see *Figure 2—figure supplement 5*). Split-half analyses of BW on cortical change (refer to *Figure 2*) showed, in general, a very low degree of replicability on the three different cortical measures. See *Supplementary file 2*. Replicability across datasets showed a similar pattern, that is, replicability was high for the effect of brain weight on cortical characteristics but very low for the effects of cortical change. See *Supplementary file 3* for stats. These analyses

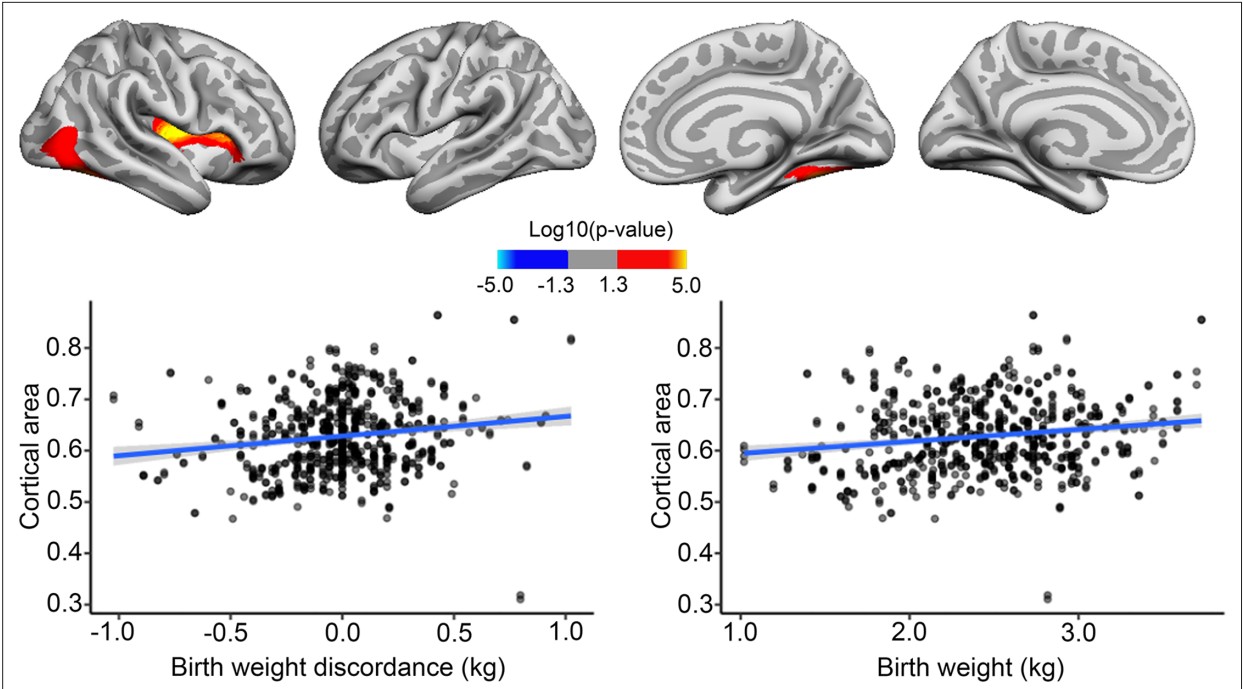

**Figure 4.** Effects of birth weight (BW) discordance on cortical area in the sample of monozygotic (MZ) twins. Significant relationships are shown from left to right: lateral view, right and left hemisphere; and medial view, right and left hemisphere. Plots are showing – for illustrative purposes – individual data points and expected trajectories for cortical area in mm (Y-axis) within the significant regions according to BW discordance (left panel) and BW (right panel) in kilograms (X-axis).

provide complementary evidence of robust associations of BW with cortical area and volume – but not cortical change – across and within samples.

## Effects of BW discordance on brain characteristics and changes in MZ twins

BW discordance analyses on twins specifically were run as described for the main analyses above, with the exception that twin scans were reconstructed using FS v6.0.1. for ABCD and the addition of the twin's mean BW as a covariate. BW discordance was associated with cortical area, where the heavier twins had greater area in some frontal, temporal, and occipitotemporal regions, with effects in the right hemisphere only surviving corrections for multiple comparisons. We note that these regions mostly overlap with regions where positive effects of BW were also seen in the bigger sample. Strikingly, the effect of BW discordance, as shown in *Figure 4*, appeared similar in size to the effect of BW itself in the MZ twin sample. However, note that this plot is merely for illustrating effects, the effect size is inflated for the BW discordance plot, since the values are derived from areas already identified as significantly related to BW. There was no association of BW discordance and cortical area changes over time.

BW discordance also had a significant negative effect on cortical thickness in restricted right frontotemporal regions, where being the lighter twin yielded greater thickness. These significant effects did not appear to overlap with regions where significant negative associations with BW were seen in the bigger sample. BW had little effect on cortical thickness in the significant region, and the effect of BW discordance in the identified regions, as shown in *Figure 5*, appeared greater than the effect of BW itself here in the MZ twin sample. However, this plot is merely for illustrating effects; it should be noted that the effect size is inflated for the BW discordance plot since the values are derived from areas already identified as significantly related to BW.

In a very small area of the right hemisphere, there was a significant association of BW discordance and cortical thickness change, meaning that the lighter twin had greater cortical thickness over time, but this effect was both regionally and quantitatively minor, as shown in *Figure 5—figure supplement 1*. There were no significant effects of BW discordance on cortical volume or volume change over time.

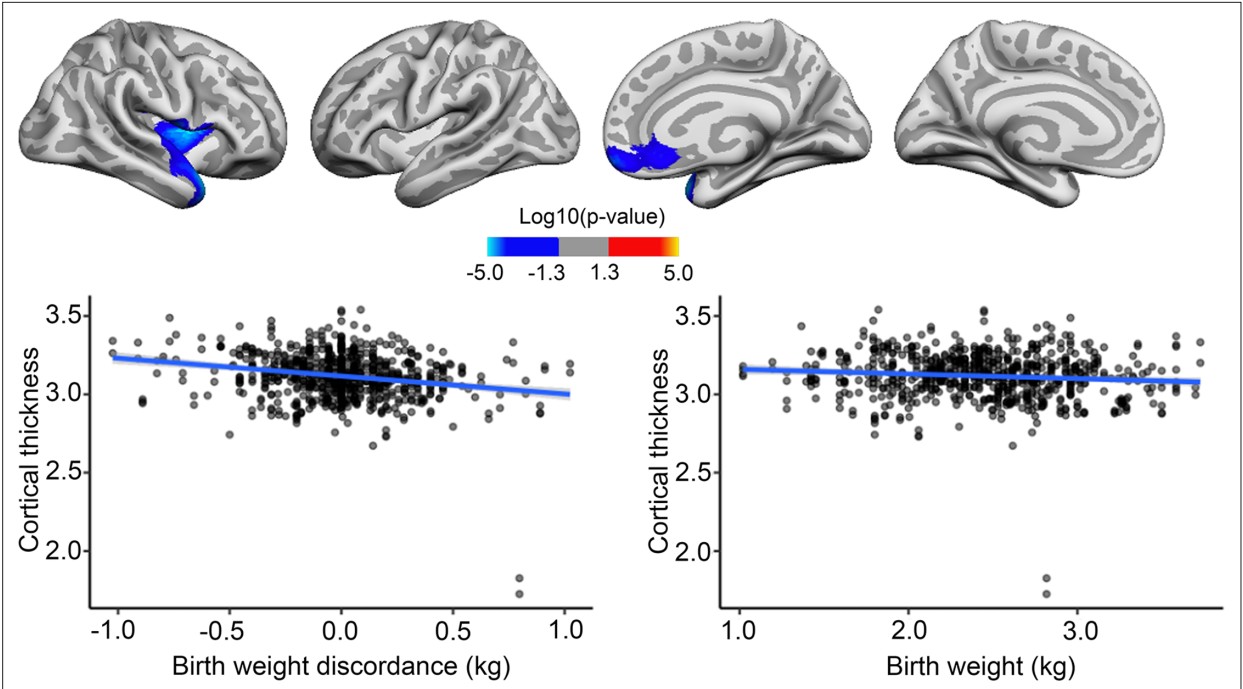

**Figure 5.** Effects of birth weight (BW) discordance on cortical thickness in the sample of monozygotic (MZ) twins. Relationships significantly corrected with cluster-forming threshold of 2.0 (p<0.01) are shown from left to right: lateral view, right and left hemisphere; and medial view, right and left hemisphere. Plots are showing – for illustrative purposes – individual data points and expected trajectories for cortical thickness in mm (Y-axis) within the significant regions according to BW discordance (left panel) and BW (right panel) in kilograms (X-axis).

The online version of this article includes the following figure supplement(s) for figure 5:

**Figure supplement 1.** Interaction effects of birth weight (BW) discordance and time on cortical thickness in the sample of monozygotic (MZ) twins.

Finally, to formally assess whether the cortical correlates (βeta-maps) of BW discordance in the twin subsample corresponded to cortical correlates of BW in the bigger samples, we did a meta-analysis of these estimates for area, thickness, and volume in the UKB, ABCD, and LCBC, and then assessed whether the cortical correlates of BW and BW discordance (βeta-maps) showed a similar topographic pattern across the datasets. The results of this meta-analysis-twin comparison showed only positive relationships, for area, $r = 0.23$, thickness $r = 0.19$, and volume $r = 0.22$. However, the respective uncorrected p-values were 0.08, 0.12, and 0.04, so the spatial comparisons would not be statistically significant (p<0.05, FDR-corrected). However, the positive correlations are suggestive that the topography of the effects of BW discordance in genetically identical twins on cortical structure was to some extent comparable to effects of individual differences in BW in the bigger sample.

**Table 1.** Descriptive statistics for the longitudinal samples.

F = number of females in the sample, M = mean, SD = standard deviation; LCBC = Lifespan Changes in Brain and Cognition; ABCD = Adolescent Brain Cognitive Development; UKB = UK Biobank. Numbers are given in years for baseline age, time since baseline, and education, birth weight is given in kilograms. For LCBC, only 584 participants had information on education. Parental education was used in ABCD, and in LCBC when the participant was below 18 y of age, and also if no other education information was available for participants up to 21 y.

| Study | N | F | Scans | Baseline age | | | Time since baseline | | | Birth weight | | | Education | | |
|---|---|---|---|---|---|---|---|---|---|---|---|---|---|---|---|
| | | | | M | SD | Range | M | SD | Range | M | SD | Range | M | SD | Range |
| LCBC | 635 | 350 | 1922 | 19.1 | 20.7 | 4.1–81.9 | 2.6 | 2.4 | 0.1–8.3 | 3.5 | 0.6 | 0.9–6.0 | 17.1 | 2.4 | 9–24 |
| ABCD | 3324 | 1562 | 6648 | 10.0 | 0.6 | 8.9–11.1 | 2.0 | 0.1 | 1.4–2.8 | 3.2 | 0.7 | 1.0–6.7 | 16.0 | 2.6 | 6–21 |
| UKB | 1759 | 1009 | 3518 | 62.0 | 7.1 | 47–80.3 | 2.3 | 0.1 | 2.0–2.9 | 3.4 | 0.6 | 0.9–6.4 | 14.2 | 2.4 | 7–16 |

## Effects of BW differences relative to other estimated effects in aging

We calculated the effect of 1 SD difference in BW (on average 600 g, see *Table 1*) on cortical and brain volume across cohorts to illustrate what variance is captured here by the early-life effects, relative to the estimates of later aging changes. The effect of 1 SD lower BW on cortical volume was 6708 mm$^3$, 8466 mm$^3$, and 5980 mm$^3$ in LCBC, UKB, and ABCD, respectively. This was equal to 1.2, 1.6, and 1.1% lesser cortex with 600–700 g lower BW for each sample, respectively. In the context of brain aging, this would be a substantial effect, higher than most risk/protective factors for dementia outlined by the Lancet Commission on dementia prevention (*Livingston et al., 2020*; *Livingston et al., 2017*), which include smoking, education, obesity, and alcohol among others (see *Walhovd et al., 2023* for a summary of the different effect sizes). The estimated yearly cortical volume reduction from 50 to 60 y is 895 mm$^3$ in the LCBC and 1402 mm$^3$ in the UKB samples, respectively. Hence, the effect of 600 g difference in BW could, if only cross-sectional data were available, be quantitatively equal to that of 7.5 and 6 y of estimated age differences in LCBC and UKB, respectively. Many factors that influence cortical volume have nothing to do with either age or differences in general brain function, including sex, so this analogy we only include to illustrate what might happen if such variance were to be ascribed to age, for example, as in modeling 'brain-age' (*Vidal-Pineiro et al., 2021*); see further discussion below.

## Discussion

The present results indicate that BW, the earliest widely and easily obtainable congenital metric, shows robust, persistent, and chiefly stable associations with brain characteristics through life. Especially, BW was associated with cortical area and volume in an age and time-invariant fashion. The robustness of this effect is quite remarkable, given the wealth of different influences individuals meet after birth, which are repeatedly assumed and reported to have a major impact on the brain through the protracted human lifespan (*Livingston et al., 2020*; *Livingston et al., 2017*). It is also in quantitative terms outstanding compared to the consistency of cortical topographies reported for other phenotypical factors (*Marek et al., 2022*). This is also special for a phenotype known to be environmentally influenced, unlike biologically hardcoded phenotypes such as sex or age, for which there are known brain-wide association studies (BWAS) patterns (*Kharabian Masouleh et al., 2019*; *Liu et al., 2020*).

Typically, other factors relating to later socioeconomic status, lifestyle, and health get the most attention in adult and aging brain research (*Livingston et al., 2020*; *Livingston et al., 2017*). Such factors, which are then targeted for prevention and intervention at different stages of the life course, often do not show consistent relationships to brain characteristics (*Walhovd et al., 2022*), may not actually be causal (*Walhovd et al., 2022*), and may themselves be related to prenatal growth (*Eriksson et al., 2000*). Another phenotype that obviously, like BW, reflects both genetic and environmental influences (*Beaumont et al., 2018*; *Li et al., 2021*; *Willis et al., 2022*) is body mass index (BMI) (*Couto Alves et al., 2019*). Consistent BWAS patterns have been reported for BMI (*Kharabian Masouleh et al., 2019*). BW stands out as the single chronological earliest phenotype, and besides BMI, BW appears to have the most replicable and consistent relations to cortical morphology, as shown here both across and within samples. It has been claimed that smaller-than-expected brain–phenotype associations and variability across population subsamples can explain widespread replication failures for BWAS (*Marek et al., 2022*). However, this is necessarily a question of which phenotypes are the most relevant to relate to brain characteristics. Also, the temporal order of factors needs to be considered if causal interpretations are to be made. Chronologically later factors necessarily do not cause earlier ones. While we cannot claim that BW itself causes the cortical characteristics observed in aging, the cortical variance explained by BW after one decade, and seven or eight decades of life alike, is unlikely to be explained by influences only present at some point in adulthood or aging. BW, as further discussed below, depends on genetic, as well as prenatal environmental influences (*Beaumont et al., 2018*; *Li et al., 2021*; *Willis et al., 2022*), which likely have causal effects on early brain morphological features. Here we find that these effects are substantial also in the aging brain.

We calculated that a BW difference of 1 SD (about 600 g) equaled a difference in cortical volume on the order of 1.1–1.6% in these cohorts. This is a quite big effect of a magnitude relevant for explaining a substantial portion of the differences typically seen between patients with neurodevelopmental or neurodegenerative diseases and healthy controls. As noted, BW differences have been

reported for neurodevelopmental disorders such as attention deficit hyperactovity disorder (ADHD) (*Momany et al., 2018*), and also other neurodevelopmental disorders, such as schizophrenia or more general psychopathology, with the cortical effects detected not invariably being very large in absolute terms (*Patel et al., 2022*). There is a limit to the range of variation that can apply to human cortical volumes, in general, (e.g., virtually none have cortical volumes below 0.45 or above 0.65 l). In terms of sample representativity, one may assume that there can be a restriction relative to the actual range of human cortical volume variation as the present samples specifically are largely healthy (*Carretta and Ree, 2022*). Much of the differences within this limited range of variation are explained by the factors we controlled for, such as age and sex. With the present effect, on the restricted range of variation, combined with big samples, it is obvious that BW differences of much less than our example magnitude (600 g) may be detectable in the cortical morphology of patients versus controls. In the context of aging and neurodegenerative change, the estimated cortical effect of ≈600 g difference in BW is of a magnitude many times the annual cortical reduction estimated to take place from, for example, 50–60 y in the adult cohorts. This is a substantial effect in brain imaging and may illuminate why metrics such as 'brain age', assumed to index aging-related processes, may rather largely capture variance already determined at birth (*Vidal-Pineiro et al., 2021*; *Franke et al., 2018*; *Wagen et al., 2022*). Neglecting this especially consistent and early factor is likely to lead to a substantial portion of human brain variance being either erroneously ascribed to factors only present at later life stages (*Vidal-Pineiro et al., 2021*; *Walhovd et al., 2023*) or left unaccounted for.

The solidity, replicability, and universality of effects as shown here for a partly environmentally influenced metric (*Beaumont et al., 2018*; *Li et al., 2021*; *Willis et al., 2022*) appear exceptional in human brain imaging. The within-sample replicability results are not fully comparable to other studies assessing the replicability of brain–phenotype associations due to analytical differences (e.g., sample size, multiple-comparison correction method) (*Marek et al., 2022*; *Kharabian Masouleh et al., 2019*). Still, these results too clearly show that the rate of replicability of BW associations with cortical area and volume is comparable to benchmark brain–phenotype associations such as age and BMI with brain structure (*Kharabian Masouleh et al., 2019*). The BW–cortical volume and area associations may be among the topographically broadest and most consistent effects so far seen as stable across the lifespan of the human brain. The three cohorts studies differ on a range of features known to be highly and reliably related to cortical characteristics, first and foremost age (*Fjell et al., 2015*; *Walhovd et al., 2016*; *Kharabian Masouleh et al., 2019*; *Kharabian Masouleh et al., 2022*; *Schnack, 2019*), but also country of origin and representativity of the populations from which they are drawn (*Garavan et al., 2018*; *Stamatakis et al., 2021*). Yet there is a comparable topography of BW effects across the samples. This is so despite the samples collectively spanning the entire human age range, within which there are always substantial age-related changes in cortical structure (*Fjell et al., 2015*; *Walhovd et al., 2016*; *Mills et al., 2016*; *Storsve et al., 2014*; *Tamnes et al., 2010*). The present results thus indicate that fetal growth influences an offset of brain reserve (*Katzman et al., 1988*; *Schofield et al., 1997*) and that this brain reserve effect is persistent and stable through the lifespan.

In contrast, the cortical maps capturing *change* in cortical structure associated with BW were not robust across datasets; that is, the most positive and negative association with BW on cortical change did not overlap at all between the different cohorts. While there was evidence from ABCD that BW affected regional cortical development in the narrow age range covered, there were limited and no consistent effects of BW on cortical change across cohorts. Importantly, there was no indication whatsoever that BW could be associated with better brain maintenance (*Nyberg et al., 2012*) in the face of age-related changes in older adulthood. Thus, the data seem to indicate that any effect of BW on cortical *change* may be of relatively more temporary nature. The 'offset effect' of BW, on the other hand, appears persistent and consistent, especially in terms of stable and widespread effects on cortical area and volume across the lifespan.

The sensitivity analyses indicate that the associations between BW and cortical characteristics are seen irrespective of not only sex and age, but also education, head size (ICV), and cases of abnormal BW. Such patterns could point to an underlying genetic pleiotropy of BW and brain characteristics. Interestingly, however, recent findings indicate that the effects of exposure to environmental adversity on epigenetic programming in aging may be localized to the in utero period (*Schmitz and Duque, 2022*). The effects of BW discordance in MZ twins in this context align with other studies (*Casey et al., 2017*; *Halevy et al., 2021*) pointing to also non-genetic, that is, environmental, influences in the

womb, associated with the pattern observed for cortical area effects. These analyses also account for multiple possibly confounding variables that could represent a mix of genetic-environmental effects, such as parental socioeconomic status, parity, or prenatal exposures shared between twins in the same womb such as maternal smoking or use of alcohol.

The neural basis for the observed association cannot readily be ascertained from human imaging studies tracking change (*Walhovd et al., 2016*; *Walhovd et al., 2012*; *Raznahan et al., 2012*). While the 'fetal origins hypothesis', proposing that cardiovascular disease in adulthood is related to under-nourishment in utero (*Barker, 1990*; *Barker, 1995*), is well known, there has been focus on 'brain-sparing' adaptations under such conditions (*Eriksson et al., 2000*). However, our finding that early human development in utero appears to be associated with a persistent and stable brain reserve effect is largely in correspondence with what is known of human nervous system development and change through the lifespan: while synaptogenesis, synaptic remodeling, and myelination are known to be protracted processes long after infancy (*Greenough et al., 1987*; *Huttenlocher, 1979*; *Petanjek et al., 2011*; *Yeung et al., 2014*), numerous processes in brain development appear to be exclusively or almost exclusively happening before birth. For instance, neurogenesis takes place almost only in fetal development (*Rakic, 1988*). Even if controversies remain, evidence suggests that any adult human neurogenesis must be severely restricted in location and amount (*Bhardwaj et al., 2006*; *Paredes et al., 2016*). Thus, human beings appear to be born with almost all cortical neurons they will have through life, and neuronal migration and differentiation are also defined early, by the place and time the neuron is born during fetal life (*Rakic, 1988*). Factors that affect placental function and uterine and/or umbilical blood flow on a chronic basis may lead to restricted fetal growth, including brain growth, and given the timing of brain development, it may not be surprising that effects would be stable across years. Animal studies of chronic placental insufficiency have shown effects on brain development that persist with age (*Rehn et al., 2004*). Hence, the relationship between BW and cortical characteristics in the normal population could likely have a twofold etiology: it is likely to in part be based on normal variation in genetically determined body and brain size, but it also may be based on variations in environmental prenatal conditions, yielding differences in optimality of early brain development persisting through the lifespan.

These results indicate that there is potential to increase brain reserve throughout the lifespan, also in aging, by combating factors affecting fetal growth negatively. While it is unknown to what extent results from these specific US and European samples can be generalized to other populations, the current potential to improve prenatal factors may be especially high in low- and middle-income countries, where the demographic changes will also be more marked in terms of the aging population (*WHO, 2019*). About 200 million children in developing countries are not meeting their growth potential, and improving the prenatal environment is likely important to help children reach their full potential (*McGovern, 2019*) – and ultimately also to help them stay above a functional threshold into older age. Also in industrialized countries, including the United States, environmental factors are associated with BW. The opioid epidemic is increasingly affecting pregnant women, and in utero opioid exposure is associated with higher risk of fetal growth restriction (*Azuine et al., 2019*). Among highly common exposures, air pollution may, for instance, be of relevance. Recently, local traffic congestion-pollution exposure during pregnancy, independently of a series of maternal sociodemographic characteristics, was associated with reductions in term BW in a large US sample (*Willis et al., 2022*). Programs and policies to limit such environmental factors reducing fetal growth may thus enhance brain reserve and ultimately prevent more people from falling below a functional threshold even in advanced age.

## Limitations

Some limitations should be noted. First, for most of the participants, only self-reported or parent-reported BW was available. While there was a very high correlation between registry and self-reported BW in LCBC, this is a possible source of noise. Second, pregnancy-related information of possible relevance, such as gestational age at birth, complications, method of delivery, maternal disorders, smoking, and alcohol and drug intake, was not available across all participants of the different samples, and was thus not analyzed here or, as for gestational age at birth, could only in part be controlled for. Some of these factors may be systematically related to BW and may thus represent confounds (*Knickmeyer et al., 2017*). There were some premature and very low BW participants in the samples, and these conditions are associated with known reductions in cortical volume (*Thompson et al., 2007*;

*Thompson et al., 2020*). However, the analyses controlling for gestational age, as well as on the restricted range of BW – excluding very preterm and very/extremely low BW children – and the analysis controlling for education, which may again relate to some of these factors, showed very similar results. It is unknown to what extent the BW of participants reflect their individual fetal growth potential as a fetus with normal BW can be growth restricted and a fetus with low BW can have appropriate growth (*Zhang et al., 2010*). We believe, however, that possible differences in such factors would likely serve to decrease consistency of results and not lead to inflated estimates of consistency.

Finally, it is beyond the scope of the present study to relate BW and cortical characteristics through the lifespan with cognitive functional differences. While all the cohorts included here also have some measures of cognitive function, they vary across samples. Furthermore, tests of cognitive function are, relative to brain imaging metrics, much more prone to test-specific test–retest effects. Thus, assessing the stability of effects across cohorts would be challenging. We note that, for example, in twins, even though there are data to suggest a relationship between BW differences and neuroanatomical features (*Raznahan et al., 2012*; *Casey et al., 2017*; *Halevy et al., 2021*), and BW differences and differences in cognitive function (*Strohmaier et al., 2015*), twins discordant for BW and neuroanatomical features may not show significant differences in neurodevelopmental outcomes (*Halevy et al., 2021*). There are many factors that influence cortical volume that have nothing to do with either age or differences in general brain function, including sex or overall differences in height. From the present data, we cannot draw conclusions about the effects on individual differences in cognitive function or its change across the lifespan. Indeed, part of BW effects on brain structure can be explained by overall somatic growth. This study does not provide one specific mechanism by which BW is associated with brain structure. Indeed, variance explained by different mechanisms may vary across samples. Further studies are needed to illuminate such questions; the present study is primarily designed to answer the question of whether associations of BW and cortical structure are consistently found throughout the lifespan.

## Conclusion

The current results show that a simple congenital marker of early developmental growth, BW, is consistently associated with lifespan brain characteristics. While some significant effects of BW on cortical

**Table 2.** MR acquisition parameters.

| Sample | Scanner | Field strength (T) | Acquisition parameters* |
|--------|---------|--------------------|-----------------------|
| LCBC | Avanto Siemens | 1.5 | TR: 2400 ms, TE: 3.61 ms, TI: 1000 ms, flip angle: 8°, slice thickness: 1.2 mm, FoV: 240 × 240 mm, 160 slices, iPat = 2 |
| | Avanto Siemens | 1.5 | TR: 2400 ms, TE = 3.79 ms, TI = 1000 ms, flip angle = 8°, slice thickness: 1.2 mm, FoV: 240 x 240 mm, 160 slices |
| | Skyra Siemens | 3.0 | TR: 2300 ms, TE: 2.98 ms, TI: 850 ms, flip angle: 8°, slice thickness: 1 mm, FoV: 256 × 256 mm, 176 slices |
| | Prisma Siemens | 3.0 | TR: 2400 ms, TE: 2.22 ms, TI: 1000 ms, flip angle: 8°, slice thickness: 0.8 mm, FoV: 240 × 256 mm, 208 slices, iPat = 2 |
| ABCD | Prisma Siemens | 3.0 | TR: 2500 ms, TE: 2.88 ms, TI: 1060 ms, flip angle: 8°, slice thickness: 1 mm, FoV: 256 × 256 mm, 176 slices, parallel imaging = 2 |
| | Achieva/ dStream/ Ingenia Phillips | 3.0 | TR: 6.31 ms, TE: 2.9 ms, TI: 1060 ms, flip angle: 8°, slice thickness: 1 mm, FoV: 256 × 240 mm, 225 slices, parallel imaging = 1.5 × 2.2 |
| | MR750/ DV25-26 GE | 3.0 | TR: 2500 ms, TE: 2 ms, TI: 1060 ms, flip angle: 8°, slice thickness: 1 mm, FoV: 256 × 256 mm, 208 slices, parallel imaging = 2× |
| UKB | Skyra Siemens | 3.0 | TR: 2000 ms, TI: 880 ms, slice thickness: 1 mm, FoV: 208 × 256 mm, 256 slices, iPat = 2 |

TR = repetition time; TE = echo time; TI = inversion time; FoV = field of view; iPat = in-plane acceleration; GRAPPA = GRAPPA acceleration factor; LCBC = Lifespan Changes in Brain and Cognition; ABCD = Adolescent Brain Cognitive Development; UKB = UK Biobank.
*Customized.

change patterns were also observed, these were regionally smaller and showed no consistency across cohorts. In conclusion, while greater early human developmental growth does not appear to promote brain maintenance in aging, it does, in terms of greater cortical volume and area, relate positively to brain reserve through the lifespan. Thus, there appears to be an omnipresence of fetal factors in the spacetime of the human brain through the lifespan. This indicates a potential to increase brain reserve at all ages, including in aging, by combating factors affecting fetal growth negatively. Given the exceptional consistency and broadness of this cortical topographical effect, it should be taken into account in studies of brain research on individual differences, whether the brains studied are those of 8- or 80-year-olds.

## Materials and methods
### Samples
In total, longitudinal data for 5718 persons with 12,088 MRI scans from the LCBC, ABCD, and UKB studies were included in the analyses. For UKB, the dataset released in February 2020 was used. For ABCD, the Data Release 3.0 was used (see http://dx.doi.org/10.15154/1528313 for this NDA (National Institutes if Health (NIH) Data Archive) study). Only persons with longitudinal MRI scans were included in the main analyses to limit the possibility that estimates of change were biased by immediate sample selection effects (i.e., those that remain for follow-up are known to have other characteristics than those who have only one time point assessment in longitudinal studies, and this can bias effects). However, for the separate MZ twin analyses, we also included participants with only one time point MRI to obtain an age-varying sample for the assessment of whether non-genetic effects were found throughout the lifespan, including in adulthood and older age. Demographics of the samples in the main analyses are given in *Table 1* and the MRI scanning parameters in *Table 2*.

### LCBC
### Population, recruitment, and general description of study/procedures
Cognitively healthy, community-dwelling participants across the lifespan were drawn from studies coordinated by the LCBC (https://www.lcbc.uio.no/english/), approved by a Norwegian Regional Committee for Medical and Health Research Ethics (REK South-East). Written informed consent was obtained from all adult participants and from parents or other legal guardians for participants below the age of majority. The samples were recruited in part by newspaper and web page ads, and in part by population registry-based research studies. Part of the developmental sample was recruited through the population registry-based study MoBa, the Norwegian Mother, Father and Child Cohort Study (MoBa) at The Norwegian Institute of Public Health (https://www.fhi.no/en/studies/moba/; *Magnus et al., 2006*, *Magnus et al., 2016*). The MoBa is a population-based pregnancy cohort study conducted by the Norwegian Institute of Public Health. Participants were recruited from all over Norway from 1999 to 2008. The establishment of MoBa and initial data collection was based on a license from the Norwegian Data Protection Agency and approval from The Regional Committees for Medical and Health Research Ethics. The MoBa cohort is currently regulated by the Norwegian Health Registry Act. The current study was approved by The Regional Committees for Medical and Health Research Ethics (REK sør-øst C 2010/2359). Part of the adult sample was recruited through the Norwegian Twin Registry (https://www.fhi.no/en/more/health-studies/norwegian-twin-registry/; *Nilsen et al., 2019*). The current NTR sub-study was approved by The Regional Committees for Medical and Health Research Ethics (REK South-East C 2018/94). Both MoBa and NTR are linked to MBRN, which is a national health registry containing information about all births in Norway. By individual consent, information on BW was also obtained from the MBRN also for some LCBC participants not being part of MoBa or NTR. Approval for these studies was given by The Regional Committees for Medical and Health Research Ethics (REK Sør-Øst B 2017/653). Most participants, including all children, were recruited for observational studies, while some adults were recruited to enter into cognitive training studies after baseline assessment (n = 168). As BW was not a criterion for assigning participants to cognitive training, these were included here.

### Inclusion/exclusion criteria/screening

Adult participants were screened using a standardized health interview prior to inclusion in the study. Participants with a history of self- or parent-reported neurological or psychiatric conditions, including clinically significant stroke, serious head injury, untreated hypertension, diabetes, and use of psycho-active drugs within the last 2 y, were excluded. Further, participants reporting worries concerning their cognitive status, including memory function, were excluded. Participants above 40 y scored $\geq 26$ on the Mini Mental State Examination (*Folstein et al., 1975*). Additionally, a generalized additive mixed model (GAMM) regressing area/thickness/volume nonlinear on age with random intercept per participant was fitted independently at each of the 163,842 vertices. Sixty observations were defined as outliers and excluded, as the absolute value of their residual was larger than four times the residual standard error at more than 6000 vertices.

### Variables used for BW and education

When possible, by participant consent, BW was obtained from the records of the MBRN (for participants born in 1967 when the registry was started or after). MBRN records of BW were obtained for a total of 526 participants. For 15 participants, BW records based on historical self-report to the Norwegian Twin Registry were obtained. Otherwise, self-report, or for children, parental report, of BW in connection with participation in the MRI scanning studies was used. For analyses controlling for gestational length, only information from the MBRN was used. Comparative analyses in the LCBC sample for 354 persons who had available both MBRN records and self-report/parent report of BW showed a very high correlation of BW as obtained from the different sources ($r = 0.99$). A high reliability of self-reported BW over time has also been found in broader NTR samples (*Nilsen et al., 2017*). For education, if multiple values were reported across time points, the highest was chosen. Education was recorded as total years of education to the highest obtained degree, for adults, and for participants <20 y of age, the average of parental education was used. However, for some participants in the age range up to 21.3 y at baseline scan, parental education was used, if no report of own education existed. Age was recorded in years and months at the time of the baseline MRI scan.

### MRI scanning and processing

MRIs were collected across two sites on 1.5T and 3T scanners (Siemens Avanto, Skyra and Prisma; Siemens Corp., Erlanger, Germany; see *Table 2*). At baseline, 345 participants were scanned at Avanto 1 at Oslo University Hospital, Oslo, 105 were scanned at Avanto 2 at St. Olav's Hospital, Trondheim, 74 were scanned at Prisma, and 111 at Skyra, Oslo University Hospital. All were followed up longitudinally at least once at the same scanner. To the extent that scanners were switched for longer follow-up, double scanning (at old and new scanner) was implemented to the extent possible, and both scans were from that time point were entered into analysis. A total of 369 participants did not switch scanner during the study, 266 switched scanner at least once, of which 106 participants were scanned with both scanners at the same time point. MRI data were processed using FreeSurfer 6.0.

## ABCD
### Population, recruitment, and general description of study/procedures

The primary aim of ABCD (https://abcdstudy.org) is to track human brain development from childhood through adolescence to determine biological and environmental factors that impact or alter developmental trajectories (*Casey et al., 2018*). ABCD has recruited >10,000 9- to 10-years-olds across 21 US sites with harmonized measures and procedures, including imaging acquisition (https://abcdstudy. org/scientists-workgroups.html). A goal of the ABCD study is that its sample should reflect, as best as possible, the sociodemographic variation of the US population (*Garavan et al., 2018*). Of relevance to the present analyses, children were ineligible to participate if they had any MRI contraindications or prematurity at birth <28 wk (*Dooley et al., 2022*; *Palmer et al., 2021*). BW was extracted from the dataset release 2.0.1 at baseline (consisting of a total of 11,875 participants), and the remainder of data were extracted from release 3.0.

### Inclusion/exclusion criteria for present analyses

All participants who had BW reported and had undergone MRI scanning at more than one time point with FreeSurfer Quality Control OK were included in the main analyses. Additionally, a GAMM regressing area/thickness/volume nonlinear on age with random intercept per participant was fitted independently at each of the 163,842 vertices. A total of 223 observations were defined as outliers and excluded as the absolute value of their residual was larger than four times the residual standard error at more than 6000 vertices.

### Variables used for BW and education

We used the fields devhx_2_birth_wt_lbs_p and devhx_2b_birth_wt_oz_p. These fields were converted to grams and added together. If devhx_2b_birth_wt_oz_p was missing, it is set to 0, so that the only contribution came from devhx_2_birth_wt_lbs_p. If devhx_2_birth_wt_lbs_p was missing, the final BW value was set to NA, and hence the participant was excluded from analysis. In cases where BW was reported multiple times for one participant and values deviated, participants were excluded from analysis if the discrepancy was greater than 10%, else the mean reported number was entered. For the analyses controlling for gestational length, we used the variable weeks_premature. Education was entered as the maximum education of the parents in years.

### MRI scanning and processing

MRIs were collected across 21 sites on 3T scanners (Siemens Prisma [Siemens Corp.], GE Discovery MR750 [GE Healthcare, Chicago, IL], and Philips Achieva [Philips, Amsterdam, the Netherlands]). The parameters are listed in *Casey et al., 2018* and at https://abcdstudy.org/images/Protocol_Imaging_Sequences.pdf. Images were processed using the FreeSurfer 7.1.0 software package for the analysis of the man longitudinal sample, whereas for the MZ twin analysis (done collectively with the UKB and LCBC MZ samples), the FreeSurfer 6.0 software package was used.

## UKB

### Population, recruitment, and general description of study/procedures

UK Biobank (UKB; https://www.ukbiobank.ac.uk/about-biobank-uk/) is a major national and international health resource with the aim of improving the prevention, diagnosis, and treatment of a wide range of illnesses. UKB recruited ≈500,000 people aged between 40 and 69 y from 2006 to 2010 from across the country to take part in this project (*Guggenheim et al., 2015*). Potential participants were identified through the National Health Service (NHS) registers according to being aged 40–69 and living within a reasonable traveling distance of an assessment center. Assessment centers (22 in total) are located in accessible and convenient locations with a large surrounding population. Participants underwent measures and provided samples and detailed information about themselves and agreed to have their health followed. Age was calculated from year and month of birth (day of month is missing, and was set to 1 for all subjects) to the date of assessment.

### Inclusion/exclusion criteria/screening

Participants were excluded from scanning in the UKB according to fairly standard MRI safety/quality criteria, such as exclusions for metal implants, recent surgery, or health conditions directly problematic for MRI scanning (e.g., problems hearing, breathing, or extreme claustrophobia) (*Miller et al., 2016*). A GAMM regressing area/thickness/volume nonlinear on age with random intercept per participant was fitted independently at each of the 163,842 vertices. Forty observations were defined as outliers and excluded as the absolute value of their residual was larger than four times the residual standard error at more than 6000 vertices.

### Variables used for BW and education

BW was extracted from the UKB data field 2022. Units are in kilogram. If multiple BWs were reported, we removed participants if the discrepancy between reported BWs was greater than 10%, and we took the mean of the reported BWs. Education: for the Biobank participants' generation, the UK school system provided free universal compulsory education between the ages of 5 and 15–16 y. Based on the UKB education data field 6138 (one college or university degree; two A levels/AS levels

or equivalent; three O levels/GCSEs (General Certificate of Secondary Education) or equivalent; four CSEs (Certificate of Secondary Education) or equivalent; five NVQ (National Vocational Qualification) or HND (Higher National Diploma) or HNC (Higher National Certificate) or equivalent; six other professional qualifications, e.g., nursing, teaching; seven none of the above; three prefer not to answer), education was recoded to years using the following dictionary: edu_ukb_to_years = (1: 16, 2: 13, 3: 11, 4: 11, 5: 11, 6: 12,–7: 10,–3: np.NaN). If multiple education values were reported, the highest education value was chosen.

## MRI scanning and processing

Imaging data were collected and processed by the UKB (https://www.ukbiobank.ac.uk) as described in *Alfaro-Almagro et al., 2018*. MRIs were collected using 3.0T Siemens Skyra (32-channel head coil). Anatomical T1-weighted magnetization-prepared rapid gradient echo images were obtained in the sagittal plane at 1 mm isotropic resolution, and T2-weighted FLAIR images were acquired at 1.05 × 1 × 1 mm resolution in the sagittal plane. Images were processed using the FreeSurfer 6.0 software package.

## Twin sample

Of the 386 MZ twins included, 310 had longitudinal imaging data. The twins were mostly (n = 310) from the developmental ABCD sample (age 10–11 y), whereas 64 adults were from the LCBC (age 18–79 y) and 12 were from the UKB (age 50–80 y).

## Statistical analyses

### Cortical vertex-wise analyses

Reconstructed cortical surfaces were smoothed with a Gaussian kernel of 15 mm full-width at half-maximum. We ran vertex-wise analyses to assess regional variation in the relationships between BW and cortical structure; area, thickness, and volume at baseline and longitudinally. In all models, we included baseline age, sex, and scanner site, as well as time (scan interval) as covariates. For ABCD specifically, ethnicity was also included as a covariate as this sample is recruited to have and has ethnic variation, whereas the other samples entered in the present analyses had little ethnic variation (i.e., in UKB, >98% of participants included in the present sample defined themselves as British/Irish/ Any other white background; in LCBC, this information was unfortunately not encoded for all, but the sample was mainly of white background). In further models, we additionally included education as a covariate. General linear models were run in turn using as predictors: BW, the interaction term birth weight × scan interval, and the interaction term of baseline age × time (scan interval) × BW. When analyses were run with baseline age × scan interval × BW as predictor, the interaction terms of baseline age × scan interval, scan interval × BW, and BW × baseline age were included as additional covariates. Standardized values were used in analyses for age, scan interval, BW, BW discordance, and education. For consistency of multiple comparison corrections across analyses, the results were thresholded at a cluster-forming threshold of 2.0, p<0.01, with a cluster-wise probability of p<0.0.25 (p<0.05/2 hemispheres). Finally, models were rerun only including participants with BWs between 2.5 and 5.0 kg to assess whether relationships were upheld also when excluding low and high BWs. Given previous findings of broad effects of BW on cortical area and volume (*Walhovd et al., 2016*; *Walhovd et al., 2012*; *Wheater et al., 2021*), we did not expect effects to be localized. Rather, we expected BW to affect gross head and brain size irrespective of sex, but we also performed supplementary analyses controlling for ICV in order to check for possible specificity of effects. Spatial correlation analyses (*Burt et al., 2020*; *Markello and Misic, 2021*; *Viladomat et al., 2014*) were run on the cortical maps (for more information, see SI) for analyses results using BW as predictor, from the LCBC, ABCD, and UKB, to assess the overlap of BW–cortical characteristics associations in terms of topography and effect sizes. In a separate set of analyses, we restricted the sample to only MZ twins, and studied effects of BW discordance (number of grams BW above or below MZ twin). In these models, we included time, baseline age, sex and site as covariates.

## Assessment of consistency of effects across and within samples

We assessed the spatial relationship between BW cortical correlates (βeta-maps) for cortical area, volume, and thickness across the different datasets (LCBC, UKB, ABCD) using Pearson's correlations. Permutation-based significance testing (n = 10.000, p<0.05, two-tailed, FDR-corrected) was performed using non-parametric spatial permutation models, that is, spin tests as implemented by *Alexander-Bloch et al., 2018*; *Lefèvre et al., 2018*. Briefly, spin tests generate spatially constrained null distributions by applying random rotations to spherical projections of the brain. For each permutation, the original values at each coordinate are replaced with those of the closest rotated coordinate. Rotations are generated in one hemisphere and mirrored in the other.

Within-sample replicability was assessed in two different ways: an exploratory and a confirmatory analysis (*Kharabian Masouleh et al., 2019*). In the exploratory analysis, we assessed within-sample replicability by conducting the same vertex-wise cortical analysis in different subsamples. For each dataset and cortical measure, we assessed the effects of BW on cortical structure and cortical change |N| = 500 times using 50% of the original sample (participants' split half). Beyond sample selection, all parameters remained identical as described in the main analysis. Vertex-wise replicability was determined by the proportion of multiple comparison-corrected results across the different subsamples. In the confirmatory analysis, we assessed the proportion of significant vertices obtained in each exploratory (train) analysis that was also deemed significant – and in the same direction – in the remaining (test; 50%) subsample (p<0.05 uncorrected). Subsamples without significant results were not considered (no results only in cortical thickness analyses). This is a criterion often followed for determining replicability (*Open Science, 2015*). Replicability analyses were performed on *fsaverage4* for computational reasons. Across-samples replicability was performed as described in the within-sample replicability analysis (i.e., we assessed the exploratory and confirmatory replicability) except that split-half was not performed – the three datasets were compared with each other – and the analyses were performed in the original *fsaverage* space.

## Acknowledgements

Data used for the LCBC sample were in part obtained through the Medical Birth Registry (MBRN) of Norway, the Norwegian Twin Registry (NTR), and the Norwegian Mother, Father and Child Cohort Study (MoBa). MoBa is supported by the Norwegian Ministry of Health and Care Services and the Ministry of Education and Research. We are grateful to all the participating families in Norway who take part in this ongoing cohort study. Data used in the preparation of this article were obtained from the Adolescent Brain Cognitive Development (ABCD) study (https://abcd-study.org), held in the NIMH Data Archive (NDA). This is a multisite, longitudinal study designed to recruit more than 10,000 children aged 9–10 y and follow them over 10 y into early adulthood. A listing of participating sites and a complete listing of the study investigators can be found at https://abcdstudy.org/consortium_members/. ABCD consortium investigators designed and implemented the study and/or provided data but did not necessarily participate in the analysis or writing of this report. This article reflects the views of the authors and may not reflect the opinions or views of the National Institutes of Health (NIH) or ABCD consortium investigators. The ABCD data used in this report came from doi:10.15154/1504041 and 10.15154/1520591. DOIs can be found at https://dx.doi.org/10.15154/1528313. Part of the research was conducted using the UK Biobank resource under application number 32048. The ERC, RCN, EC, NIH, and UK Biobank had no role in the design and conduct of this specific study. We thank all participants for contributing to all the respective data sources. This study was funded by ERC grants (771375 and 313440 to KBW; 283634 and 725025 to AMF), Research Council of Norway (RCN) grants (to KBW, AMF, and DVP), and European Commission (EC) EU Horizon 2020 Grant agreement number 732592 (Lifebrain). The ABCD Study is supported by the National Institutes of Health and additional federal partners under award numbers U01DA041048, U01DA050989, U01DA051016, U01DA041022, U01DA051018, U01DA051037, U01DA050987, U01DA041174, U01DA041106, U01DA041117, U01DA041028, U01DA041134, U01DA050988, U01DA051039, U01DA041156, U01DA041025, U01DA041120, U01DA051038, U01DA041148, U01DA041093, U01DA041089, U24DA041123, and U24DA041147. A full list of supporters is available at https://abcdstudy.org/federal-partners.html.

# Additional information

## Funding

| Funder | Grant reference number | Author |
|---|---|---|
| European Research Council | 771375 | Kristine B Walhovd |
| European Research Council | 313440 | Kristine B Walhovd |
| European Research Council | 283634 | Anders Martin Fjell |
| European Research Council | 725025 | Anders Martin Fjell |
| H2020 European Research Council | 732592 | Kristine B Walhovd |
| Norwegian Research Council | 324882 | Didac Vidal-Pineiro |

The funders had no role in study design, data collection and interpretation, or the decision to submit the work for publication.

## Author contributions

Kristine B Walhovd, Conceptualization, Resources, Data curation, Formal analysis, Supervision, Funding acquisition, Investigation, Methodology, Writing - original draft, Project administration, Writing – review and editing; Stine K Krogsrud, Investigation, Project administration, Writing – review and editing; Inge K Amlien, Data curation, Formal analysis, Investigation, Methodology, Writing – review and editing; Øystein Sørensen, Fredrik Magnussen, Formal analysis, Methodology, Writing – review and editing; Yunpeng Wang, Anne Cecilie S Bråthen, Knut Overbye, Jonas Kransberg, Asta K Håberg, Investigation, Writing – review and editing; Athanasia M Mowinckel, Data curation, Investigation, Writing – review and editing; Martine Herud, Data curation, Writing – review and editing; Anders Martin Fjell, Conceptualization, Funding acquisition, Investigation, Methodology, Writing – review and editing; Didac Vidal-Pineiro, Conceptualization, Data curation, Formal analysis, Investigation, Methodology, Writing – review and editing

## Author ORCIDs

Kristine B Walhovd ⓘ http://orcid.org/0000-0003-1918-1123
Athanasia M Mowinckel ⓘ https://orcid.org/0000-0002-5756-0223
Fredrik Magnussen ⓘ http://orcid.org/0000-0003-2574-1705
Asta K Håberg ⓘ http://orcid.org/0000-0002-9007-1202
Anders Martin Fjell ⓘ http://orcid.org/0000-0003-2502-8774
Didac Vidal-Pineiro ⓘ http://orcid.org/0000-0001-9997-9156

## Ethics

The study was, approved by a Norwegian Regional Committee for Medical and Health Research Ethics (REK South-East). For LCBC, written informed consent was obtained from all adult participants and from parents or other legal guardians for participants below age of majority. The establishment of MoBa and initial data collection was based on a license from the Norwegian Data Protection Agency and approval from The Regional Committees for Medical and Health Research Ethics. The MoBa cohort is currently regulated by the Norwegian Health Registry Act. The current study was approved by The Regional Committees for Medical and Health Research Ethics (REK sU00F8;r-x00F8;st C 2010/2359). See https://www.ukbiobank.ac.uk/about-biobank-uk/ and https://abcdstudy.org for specific details for details on specific ethics and informed consents.

Reviewer #1 (Public Review): https://doi.org/10.7554/eLife.86812.3.sa1
Reviewer #2 (Public Review): https://doi.org/10.7554/eLife.86812.3.sa2
Author response https://doi.org/10.7554/eLife.86812.3.sa3

# Additional files

## Supplementary files

- MDAR checklist

- Supplementary file 1. Spatial correlation of birth weight effects on brain structure across datasets.

- Supplementary file 2. Exploratory and confirmatory replicability of birth weight on cortical change within datasets.

- Supplementary file 3. Exploratory and confirmatory replicability across datasets.

## Data availability

The code used for vertex-wise ST-LME analyses and spatial correlations analyses can be accessed here https://github.com/LCBC-UiO/paper-birthweight-brainchange-2022 (copy archived at *Vidal-Pineiro and Amlien, 2024*). Two of the datasets used, the ABCD https://abcdstudy.org and UKB https://www.ukbiobank.ac.uk databases are open to all researchers given appropriate application, please see instructions for how to gain access here: https://nda.nih.gov/abcd/ and https://www.ukbiobank.ac.uk/enable-your-research. The LCBC dataset has restricted access, requests can be made to the corresponding author, and some of the data can be made available given appropriate ethical and data protection approvals. However, the registry data on birth weight connected to this sample are not shareable by the authors, as these data are owned by the Medical Birth Registry of Norway, https://www.fhi.no/en/hn/health-registries/medical-birth-registry-of-norway/medical-birth-registry-of-norway/, the Norwegian Mother, Father, and Child Cohort Study https://www.fhi.no/en/studies/moba/, and the Norwegian Twin Registry https://www.fhi.no/en/more/health-studies/norwegian-twin-registry/ so that any access to data must be approved by them. Group-level unthresholded p-maps, F-maps, Beta-maps, and degrees of freedom for the univariate analyses accompany this manuscript as additional material.

The following previously published datasets were used:

| Author(s) | Year | Dataset title | Dataset URL | Database and Identifier |
|---|---|---|---|---|
| Jernigan TL, Brown SA, Dale AM, Tapert SF, Sowell ER, Herting M, Laird A, Gonzalez R, Squeglia L, Gray K, Paulus MP, Aupperle R, Feldstein Ewing SW, Nagel BJ, Fair DA, Baker F, Colrain IM, Bookheimer SY, Dapretto M, Brown SA, Jacobus J, Hewitt JK, Banich MT, CottlerLB , Nixon SJ, Ernst TM, Chang L, Heitzeg MM, Sripada C, Luciana MM, Iacono WG, Clark DB, Luna B, Foxe J, Freedman E, Yurgelun-Todd DA, Renshaw PF, Potter A, Garavan HP, Lisdahl K, Larson C, Bjork JM, Neale MC, Heath AC, Barch DM, Madden PA, Casey BJ, Baskin-Sommers A, Gee D | 2017 | Adolescent Brain Cognitive Development DEAP Study (ABCD) release 3.0 #1042 | https://doi.org/10.15154/1520591 | NIMH Data Archive, 10.15154/1520591 |

*Continued on next page*

*Continued*

| Author(s) | Year | Dataset title | Dataset URL | Database and Identifier |
|---|---|---|---|---|
| Yang R, Jernigan T, Casey B, Clark D, Colrain I, Dale A, Ernst T, Gonzalez R, Heitzeg M, Lisdahl K, Luciana M, Nagel B, Sowell E, Squeglia L, Tapert S, Yurgeluntodd D | 2017 | Adolescent Brain Cognitive Development Study (ABCD) 2.0.1 release #721 | https://doi.org/10.15154/1504041 | NIMH Data Archive, 10.15154/1504041 |

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
