## [Editor Report · eLife assessment]

This **valuable** study uses multiple large neuroimaging datasets acquired at different points through the lifespan to provide **solid** evidence that birth weight (BW) is associated with robust and persistent variations in cortical anatomy, but less-substantial influences on cortical change over time. These findings, supported by robust statistical methods, illustrate the long temporal reach of early developmental influences and carry relevance for how we conceptualize, study, and potentially modify such influences more generally. The article will be of interest to people interested in brain development and aging.

---

## [Referee Report · Reviewer #1 (Public Review)]

This manuscript uses 3 large neuroimaging datasets - which together span childhood to late adulthood - to model the relationship between birthweight (BW) and cortical anatomy over time. The authors separately consider BW associations with the "height" of cortical anatomy trajectories (intercept effects) vs. BW associations with trajectory shape. They authors also distinguish between BW associations with cortical surface area (SA) and cortical thickness (CT), which together determine cortical volume (CV). Prior studies have firmly established robust positive associations between BW and cortical SA, but this study adds evidence for the protracted lifespan persistence of these associations, and the degree to which BW associations with cortical change over time are much weaker.

The study has several strengths including: clearly motivation of this work in the Introduction and contextualization of the results in Discussion; use of three large neuroimaging datasets; inclusion of sensible sensitivity analyses; disambiguation of SA and CT findings; and use of formal spatial analysis to quantify the reproducibility of effects across cohorts.

The primary way in which this work seeks to extend beyond established findings is to determine if BW is associated with differences in cortical change over time. The results presented clearly establish that such BW-change associations are much more localized and less consistent across cohorts that BW-intercept associations. The authors use multiple complementary approaches to verify the robustness of this inference to dataset subsampling and variation in statistical methods.

Overall, this work provides a valuable new data point in our understanding of the profound and protracted influences that prenatal developmental features can have on postnatal outcomes.

---

## [Referee Report · Reviewer #2 (Public Review)]

This study focuses on the association between weight at birth and area, volume and thickness of the cerebral cortex measured at timepoints throughout the lifespan. Overall, the study is well designed, supported by evidence from a large sample drawn from three geographically distinct cohorts with robust analytical and statistical methods.

The authors test the hypotheses: that higher birth weight is associated with greater cortical area in later life; that associations are robust across samples and age; and that associations are stable across the lifespan. Analyses are performed separately in three cohorts: ABCD, UKBB and LCBC and the pattern of associations compared by means of spatial correlations. They find that BW is positively associated with cortical area (and, as a consequence, cortical volume) across most of the cortex, with effect sizes greatest in frontal and temporal regions. These associations remain largely unchanged when accounting for age, sex, length of gestation and (in one cohort) ethnicity. Variations due to MRI scanner and site are accounted for statistically. Measures are taken to determine within sample replicability through split-half analyses.

The authors conclude that BW, as a marker of early development, is associated with brain characteristics throughout the lifespan.

---

## [Author Response]

The following is the authors’ response to the original reviews.

**Reviewer #1**

Reviewer #1’s main concerns revolved around the evidential strength of the study’s conclusion that age-specific effects of birth weight on brain structure are more localized and less consistent across cohorts than age-uniform, stable effects. Specifically, the reviewer points out the evidence (or lack of such) for age-specific effects. We have rearticulated as a “bullet-point summarization” the reviewer’s concerns for a better response (please, see the original reviewer’s response in the annexed document). We thank the reviewer for his/her comment.

Concern #1: No direct statistical comparisons are conducted between samples (beyond the spin-tests).

In the initial version of the manuscript, the spin-tests represented a key test since they compared the spatial distribution of birth weight effects across samples. In the revised manuscript, we additionally perform a replicability analysis across samples both for birth weight effects on brain characteristics and on brain change in a similar fashion as described for the within-sample analysis. The results of these analyses provide complementary evidence of robust associations of birth weight effects on cortical characteristics (for area and volume, less so for thickness) and of unreliable associations of birth weight on cortical change. These analyses are briefly mentioned in the main document and fully described as supplementary information. Briefly, the effects of birth weight on cortical area and cortical volume showed high (exploratory and confirmatory) replicability while replicability was almost nonexistent for the effects of birth weight on cortical change. See below, under Reviewer #1, concern #2, for a description of the changes in the revised manuscript.

Concern #2: The differential composition of samples in terms of age distribution leads to the possibility that lack of results is explained by methodological differences.

The revised version of the manuscript provides now a within-sample replicability analysis of the birth weight effects on cortical change. This analysis addresses the reviewer’s concern as the lack of replicability in this analysis cannot be attributed to sample or methodological differences. We thank the reviewer for suggesting this analysis which provides further quantification of the (lack of) robustness of the birth weight effects on cortical change. See below for changes in the revised version of the manuscript concerning additional replicability analyses which were carried out as a response to reviewer #1 concerns #1 and #2.

pp. 12-3. “Additionally, we performed replicability analyses both across and within samples to further investigate the robustness of the effects of birth weight on cortical characteristics and cortical change. Split-half analyses within datasets were performed, to investigate the replicability of significant effects 36,37 of BW on cortical characteristics within samples (refer to Figure 1). These analyses further confirmed that the significant effects were largely replicable for volume and area, but not for thickness (see Supplementary Figure 11). Split-half analyses of BW on cortical change (refer to Figure 2) showed, in general, a very low degree of replicability on the three different cortical measures. See Supplementary Table 3. Replicability across datasets showed a similar pattern, that is, replicability was high for the effect of brain weight on cortical characteristics but very low for the effects of cortical change. See Supplementary Table 4 for stats. See Supplementary statistical methods for a full description of the analyses. These analyses provide complementary evidence of robust associations of BW with cortical area and volume – but not cortical change - across and within samples.”

p. 41. “For each dataset and cortical measure, we assessed the effects of birth weight on cortical structure and cortical change (…)”

p. 42. “Across samples replicability was performed as described in the within-sample replicability analysis (i.e., we assessed the exploratory and confirmatory replicability) except that split-half was not performed - the three datasets were compared with each other - and the analyses were performed in the original fsaverage space.”

pp. 54-55. “The exploratory replicability of birth weight on cortical change was negligible across datasets and measures [.00 (.00), .00 (.00), .00 (.00) for area, .02 (.09), .00 (.02), .01 (.03) for volume, and .01 (.05), .01 (.14), .00 (.01) for thickness] while confirmatory replicability was generally poor, except for the ABCD dataset [.02 (.05), .68 (.35), .00 (.00) for area, .08 (.14), .56 (.25), .00 (.02) for volume, and .37 (.26), .60 (.27), .01 (.03) for thickness] (see Supplementary Table 3).

These results are not fully comparable to other studies assessing the replicability of brain phenotype associations due to analytical differences (e.g. sample size, multiple-comparison correction method)20,36, yet clearly show that the rate of replicability of BW associations with cortical area and volume are comparable to benchmark brain-phenotype associations such as body-mass index and age68. Lower levels of replicability in the LCBC subsample are likely attributable to higher sample variability (e.g. increased age span). Kinship may lead to inflated patterns of replicability within the ABCD cohort. Confirmatory replicability is, also, to some degree, affected by sample size, and thus the estimates of confirmatory replicability may be somewhat inflated in the ABCD dataset.

Finally, the degree of across-sample replicability was high for the effects of birth weight on cortical area and volume (average confirmatory replicability = .96 and .93), low for thickness (.27), and negligible for the effects of birth weight on cortical change (.03, .06, and .06). See further information in Supplementary Table 4.”

Concern #3: Some datasets have a narrow age range precluding the detection of age-related effects.

We do not believe concern #3 is a major problem since timebirth weight refers to a within subject contrast, e.g., longitudinal-only-based contrast. Birth weight, even when self reported, is a highly reliable measure and the sample sizes are relatively large (n = 635, 1759, and 3324 unique individuals). Note that the smaller dataset does have longer follow-up times and more observations per participant, increasing the reliability of estimations in individual change. Structural MRI measures have very high reliability. Clearly, longitudinal brain change is less reliable, yet the present sample size and the high reliability of birth weight should provide enough statistical power to capture even small time-varying effects of birth weight on brain structure. Note as well that in each model age is treated as a covariate. Rather, the consistency of timebirth weight (that is, the effects of birth weight on cortical change) is assessed with split-half replications within and across samples. In this methodological pipeline, a narrow age range for a given dataset, if anything, may constitute an advantage. We have clarified the statistical model (see changes in the revised manuscript, referred to in response to reviewer #1, concern #5).

Concern #4 The modeling strategy does not allow for non-linear interaction between age and BW suggesting the use of spline models instead in a mega-analytical fashion.

Indeed, we agree that some - if not most - brain structures follow non-linear trajectories throughout life. In the present study, age regressors are used only for accounting for variance in the data rather than capturing any effect of interest. Rather, it is the time*birth weight regressor that captures age-varying changes in brain structure. Time reflects within-subject follow-up time. We believe non-linear modeling of age will only account for additional variance (compared to linear models) in the LCBC dataset given the dataset’s wider age range, while it will not have any consequential effect in the ABCD and UKB datasets (as predicted in the provisional response). In any case, we recognize it as a valid concern. Consequently, we have rerun the main models in an ROI-based fashion using or not using spline models to fit age. Specifically, we have fitted the models in each of Desikan-Killiany’s ROIs using generalized additive mixed models (GAMM with age as a smooth term) or linear mixed models (LME with age as a linear regressor). The results are shown in Supplementary Figures 13 and 14. The Beta regressors are nearly identical. As expected, the differences are noticeable in the LCBC dataset while the effect of using - or not using- splines to fit age is almost null in the other two datasets. See also FDR-corrected maps below for both birth weight effects on brain structure and brain change (we opted to show Beta-maps as supplementary material as the multiple-comparisons correction in the ROI-based analysis is not fully comparable with the one used in the vertex-wise approach).

p. 9: “Both birth weight effects on cortical characteristics and cortical change were rerun (ROIwise) using spline models that accounted for possible non-linear effects of age on cortical structure. The results were comparable to those reported above in Figures 1 and 2. See Supplementary Figures 13 and 14 for birth weight effects on cortical characteristics and cortical change, respectively.”

Caption to Supplementary Figure 13. “Comparison between spline (GAMM) and linear (LME) models on the effect of birth weight on cortical characteristics. Age was fitted either as a smoothing spline using generalized additive mixed models (GAMM, mgcv r-package) or a linear regressor with a linear mixed models (LME, lmer r-package) framework. The analyses were performed ROI-wise using the Desikan-Killiany atlas. Significance was considered at a FDR corrected threshold of p < 0.04. All the remaining parameters were comparable to the main analyses shown in Figure 1. The viridis-yellow scale represents the lower-higher Beta regressors. Red contour displays regions showing significant effects of birth weight. Note the high correspondence with both fitting models. Differences are only noticeable in the LCBC sample due to the datasets’ wider age range (i.e., lifespan dataset).”Caption to Supplementary Figure 14. “Comparison between spline (GAMM) and linear (LME) models on the effect of birth weight on cortical change. Age was fitted either as a smoothing spline using generalized additive mixed models (GAMM, mgcv r-package) or a linear regressor with a linear mixed models (LME, lmer r-package) framework. The analyses were performed on ROI-based using the Desikan-Killiany atlas. Significance was considered at a FDR corrected threshold of p < 0.04. All the remaining parameters were comparable to the main analyses shown in Figure 1. The viridis-yellow scale represents the lower-higher Beta regressors. Red contour displays regions showing significant effects of birth weight. Note the high correspondence with both fitting models. Differences are only noticeable in the LCBC sample due to the datasets’ wider age range (i.e., lifespan dataset).”The figures below show the birth weight effects on brain characteristics (above) and change (below) using a GAMM or an LME approach; that is, using age as a smooth term or as a regressor. FDR-corrected p < 0.05 values are shown in a signed logarithmic scale. Red-yellow values represent positive associations between birth weight and brain while blue-lightblue values represent negative associations. The results are qualitatively comparable and quantitative differences exist only in the LCBC dataset. Please see Supplementary Figures 13 and 14 in the revised manuscript.

Concern #5: Greater clarity regarding the statistical models and the provision of effect-size maps.

The revised manuscript provides additional information regarding the statistical model, especially in the results section, to avoid misunderstanding (see below examples of clarifications in the revised manuscript). We now provide Beta-maps, F-maps, unthresholded p-values maps, and degrees of freedom for the main univariate analyses. That is, we provide this information for both the whole sample and the twin analyses which correspond to Figures 1, 2, 4, and 5. We opted not to compute effect-size estimates (e.g. partial eta-squared, cohen’s d) due to the ambiguous interpretation of these maps in the context of linear mixed models.

p.8. “To test the effect of birth weight on cortical change we rerun the analyses with BW x time and age x time interactions. Note BW x time (i.e., within-subject follow-up time) represents the contrasts of interest while age – and age interactions – are used to account for differences in age across individuals.”

p.11. “In contrast, the spatial correlation of the maps capturing BW-associated cortical change (i.e., BW x time contrast) …”

p. 12. “Additionally, we performed replicability analysis both across and within samples to further investigate the robustness of the effects of birth weight on cortical characteristics and cortical change.”

p. 14: “BW discordance analyses on twins specifically were run as described for the main analyses above, with the exception that twin scans were reconstructed using FS v6.0.1. for ABCD and the addition of the twin’s mean birth weight as a covariate.”

p .31. “Group-level unthresholded p-maps, F-maps, Beta-maps, and degrees of freedom for the univariate analyses accompany this manuscript as additional material.”